# Bioinspired 3D structures with programmable morphologies and motions

Amirali Nojoomi[1], Hakan Arslan[2], Kwan Lee[1] & Kyungsuk Yum[1]

Living organisms use spatially controlled expansion and contraction of soft tissues to achieve complex three-dimensional (3D) morphologies and movements and thereby functions. However, replicating such features in man-made materials remains a challenge. Here we report an approach that encodes 2D hydrogels with spatially and temporally controlled growth (expansion and contraction) to create 3D structures with programmed morphologies and motions. This approach uses temperature-responsive hydrogels with locally programmable degrees and rates of swelling and shrinking. This method simultaneously prints multiple 3D structures with custom design from a single precursor in a one-step process within 60 s. We suggest simple yet versatile design rules for creating complex 3D structures and a theoretical model for predicting their motions. We reveal that the spatially nonuniform rates of swelling and shrinking of growth-induced 3D structures determine their dynamic shape changes. We demonstrate shape-morphing 3D structures with diverse morphologies, including bioinspired structures with programmed sequential motions.

[1] Department of Materials Science and Engineering, University of Texas at Arlington, 501 West First Street, Arlington, TX 76019, USA. [2] Department of Mechanical and Aerospace Engineering, University of Texas at Arlington, 500 West First Street, Arlington, TX 76019, USA. Correspondence and requests for materials should be addressed to K.Y. (email: kyum@uta.edu)

Nature has inspired researchers to develop shape-morphing materials that can replicate the functions of native soft tissues[1,2]. Such materials have applications in soft robotics, programmable matter, bioinspired engineering, and biomimetic manufacturing[1–12]. Existing approaches use swellable hydrogels[3–8,13], shape-memory polymers[14,15], and liquid crystalline elastomers[16–18] with fabrication methods, such as photopatterning[4,6–8,11], self-folding[7,11,19,20], and three-dimensional (3D) printing[5,15]. These approaches have been used to build various self-shaping 3D structures, including those with complex 3D morphologies of living organisms[4,5], but reproducing their movements has not been fully achieved[1,2,21].

A promising approach in this regard is to use spatially controlled in-plane growth (expansion and contraction) of hydrogel sheets to form 3D structures via out-of-plane deformation (non-Euclidean plates)[3,4]. Because bending is energetically less expensive than stretching in a thin sheet, the internal stresses developed by nonuniform in-plane growth are released by out-of-plane deformation[3,4]. This approach defines 3D shapes with Gaussian curvatures[22,23] and is uniquely capable of creating 3D structures with curved geometries, often seen in biological organisms but difficult to achieve by other methods[4,5]. Living organisms, ranging from plants to marine invertebrates, use such approaches (e.g., differential growth) for fundamental biological processes, including morphogenesis, complex growth and movement, and adaptation to environments[3–5,13,24–34]. With the physical properties of hydrogels similar to those of soft tissues[7,12,33], this approach thus has great potential for creating bioinspired 3D structures[4]. In particular, the ability to spatially and temporally control the local in-plane growth could offer a new strategy to create dynamic 3D structures that can mimic the continuously deforming motions of living organisms[1,2]. However, dynamic growth-induced 3D motions of non-Euclidean plates remain largely unexplored[3,4]. Previous theoretical and experimental studies have mainly focused on the formation of 3D shapes at equilibrium states[3,4,35–37], but their dynamic behavior at metastable states during shape transition is not well understood.

Furthermore, the principle has been demonstrated for various 3D shapes[3,4,37], but achieving nonaxisymmetric 3D structures with complex morphologies remains to be further studied[34–36].

Here we show an approach named digital light 4D printing (DL4P) that creates dynamic 3D structures with programmed morphologies and motions (Fig. 1a). This approach encodes temperature-responsive 2D hydrogels with spatially and temporally controlled growth (expansion and contraction) functions $\Omega$, or target metrics, which transforms the hydrogels into prescribed 3D structures and programs their motions. Previous studies of differential growth-induced 3D shaping have mostly formed single 3D shapes either at the swelled or the shrunk state[3–5,38]. In contrast, our temperature-responsive hydrogels with phototunable degrees and rates of swelling and shrinking allow us to define target 3D shapes at both the swelled and shrunk states. In particular, the ability to control the rates uniquely enables a new strategy for programming growth-induced 3D motions. This method simultaneously prints multiple 3D structures with custom design (using digital light projection grayscale lithography) from a single precursor solution in a one-step process within 60 s and is thus highly scalable. Taking advantage of our phototunable hydrogels and the flexible 2D printing method for 3D material programming (e.g., without the need for multiple physical masks or nozzles), we established simple yet versatile design rules and the concept of modularity for creating complex 3D structures with diverse morphologies[4], including ray-inspired structures with programmed motions. To investigate the dynamic growth-induced motions, we introduced a concept of dynamic target metrics and developed a dynamic theoretical model based on the concept. Our experimental and theoretical studies reveal that the spatially nonuniform rates of swelling and shrinking of growth-induced 3D structures determine their dynamic shape changes. Furthermore, the swelling and shrinking rates of our hydrogels are phototunable and thus locally programmable. The ability to spatially control the rates of shape changes allows us to fabricate dynamic 3D structures with programmed sequential motions, as previously demonstrated with photopatterned hydrogels

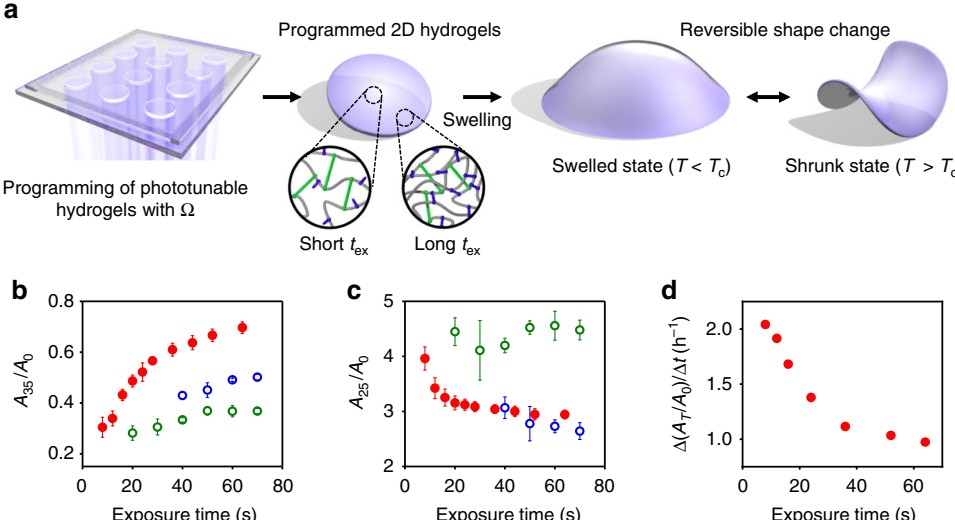

**Fig. 1** Programming of phototunable hydrogels to create 3D structures. **a** Digital light 4D printing process. The hydrogels are encoded with a growth function (or target metric) $\Omega$ using digital light projection grayscale lithography. The created 3D structures undergo a reversible shape transition at volume phase transition temperature $T_c$ (~32.5 °C). $T$ is temperature. The insets illustrate the polymer networks of the hydrogels at the early (short light exposure time $t_{ex}$) and late (long $t_{ex}$) stages of photopolymerization, where gray, blue, and green structures represent pNIPAm, BIS, and PEGDA, respectively. **b, c** Areal shrinking (**b**) and swelling (**c**) ratios of dual- (red circles), BIS- (blue open circles), and PEGDA (green open circles)-crosslinked pNIPAm hydrogels as a function of $t_{ex}$. $A_{35}$ and $A_{25}$ are the areas of hydrogels at 25 °C and 35 °C, respectively. $A_0$ is the area of as-prepared hydrogels. Error bars: s. d. of three independent measurements. **d** Areal swelling rates $\Delta(A_T/A_0)/\Delta t$ of dual-crosslinked pNIPAm hydrogels as a function of $t_{ex}$. $A_T$ is the area of hydrogels. $t$ is time

responsive to different molecular inputs (e.g., DNA molecules with different sequences) and those with different thicknesses[8]. Such ability is critical for implementing complex functions but challenging to attain with global external stimuli (e.g., temperature)[8]. This work introduces a 3D fabrication method with the advantages of traditional (scalable) and additive (customizable) manufacturing for fabricating soft devices with programmed 3D morphologies and motions.

## Results

**Hydrogels with phototunable material properties.** Our DL4P approach relies on the ability to prepare temperature-responsive hydrogels with continuously varying compositions and thus material properties (e.g., degrees and rates of swelling and shrinking) from a single precursor solution through photo-polymerization and crosslinking within 60 s (Fig. 1). The modulation of the material properties is based on the temporal control of polymerization and crosslinking reactions using two types of crosslinkers with different lengths by light exposure time $t_{ex}$. The phototunability provides a flexible means to encode the hydrogels with spatially and temporally controlled growth (swelling and shrinking), which can be used to program the formation of 3D structures and their motions.

The precursor solution consists of $N$-isopropylacrylamide (NIPAm), $N,N'$-methylene bisacrylamide (BIS; short-chain crosslinker), and poly(ethylene glycol) diacrylate (PEGDA; long-chain crosslinker). For an equimolar concentration of crosslinkers, crosslinking with PEGDA forms gels faster than with BIS, due to longer distances between the crosslinking points of PEGDA (Supplementary Figures 1, 2)[39]. The BIS- and PEGDA-crosslinked hydrogels swell and shrink in different degrees (Fig. 1b, c, Supplementary Figure 1).

Based on these results, we hypothesized that poly($N$-isopropylacrylamide) (pNIPAm) hydrogels crosslinked with both BIS and PEGDA have a larger phototunable range of swelling and shrinkage over a wider range of $t_{ex}$ than those crosslinked with single crosslinkers. The dual crosslinking indeed increases the phototunable range of shrinking and swelling and the range of $t_{ex}$ that can be used to tune the shrinking and swelling ratios (Fig. 1b, c, Supplementary Figure 3). More interestingly, the swelling and shrinking rates of our hydrogels are also phototunable (Fig. 1d). We reason that crosslinking with long-chain crosslinkers (PEGDA) forms a low density hydrogel framework at an early stage (low monomer conversion), whereas the conversion of monomers to polymers and their crosslinking via short-chain crosslinkers (BIS) continuously occurs within the hydrogel framework throughout the time course of photopolymerization, increasing the density of the polymer networks (Fig. 1a). Moreover, crosslinking via PEGDA is expected to be suppressed at the late stage, because of diffusional limitations in high-density polymer networks[39]. We verified this mechanism by measuring the density of the polymer networks as a function of $t_{ex}$. The density increases with $t_{ex}$ (Supplementary Figure 4a). The increase in the density in turn reduces the degrees and rates of macroscopic swelling and shrinking (Fig. 1b–d, Supplementary Figure 4b, c). This mechanism differs from previous ones that control the crosslink density by light irradiation dose, which, for example, tunes only the swelling of pNIPAm hydrogels[4]. In contrast to photoinduced controlled/living radical polymerization (photo-CRP), which precisely controls the molecular architecture of polymers, such as molecular weights and compositions[40–43], our approach modulates only the overall density of polymer networks (rather than the molecular weight of individual polymer chains).

**Shape-morphing 3D structures with axisymmetric metrics.** To validate our DL4P approach and demonstrate its accuracy, we created well-defined geometric 3D structures with axisymmetric metrics (Fig. 2a–k). In contrast to previous studies, which mostly form 3D shapes at either the swelled or the shrunk state[3–5], our approach can define the target 3D shapes at both the swelled ($\Omega > 1$) and the shrunk ($\Omega < 1$) states. The equilibrium 3D shape is selected from the competition between bending ($E_B \sim t_h^3$, where $t_h$ is the thickness of a sheet) and stretching ($E_S \sim t_h$) energies[3,38]. As the thickness decreases, the hydrogel sheet thus converges to the stretch-free configuration that fully follows the target metric[38]. However, the actual metric adopted by experimental 3D structures differs from the target metric, because of a finite-thickness bending energy[36,38]. The structure at the shrunk state can thus yield a 3D shape closer to the theoretical configuration described by the target metric than one at the swelled state. In addition, the use of hydrogels at the shrunk state is beneficial for practical applications, for example, because of their enhanced mechanical properties and the formation of target shapes under physiological conditions ($T = 37\,°C$) for potential biomedical applications[44]. We thus designed $\Omega$ for target shapes at the shrunk state.

We created spherical cap, saddle, and cone structures with constant Gaussian curvature $K > 0$, $K < 0$, and $K = 0$, respectively (Fig. 2a–c). We formed these structures by encoding hydrogels (400 μm in thickness) with $\Omega$ shown in Fig. 2g (see Supplementary Notes 1–3 for the theoretical model). The resulting structures agree quantitatively with the theoretical structures, reflecting the accuracy of our approach (Fig. 2d–f, Supplementary Note 3). For example, the experimentally measured $K$ of the spherical cap and saddle structures are $0.0464\,\text{mm}^{-2}$ and $-0.0727\,\text{mm}^{-2}$, which match well with the theoretically calculated $K$ of $0.0468\,\text{mm}^{-2}$ and $-0.0722\,\text{mm}^{-2}$, respectively. The cone structures constructed with different exponents $\alpha$ in $\Omega$ have the programmed value of the vertex angle $\beta$ ($\beta = \sin^{-1}\alpha$) (Fig. 2h, Supplementary Note 3). We further verified our approach by creating Enneper's minimal surfaces (Fig. 2i, j) using $\Omega(r) = c[1 + (r/R)^{2(n'-1)}]^2$, where $r$ is a radial position and $c$, $R$, and $n'$ are constants (Fig. 2k)[4]. As expected, the growth functions with different $n'$ induce Enneper's surfaces with the targeted number of wrinkles $n'$ (Fig. 2i–k, Supplementary Figure 5, Supplementary Movie 1).

The created structures reversibly transform between prescribed 3D shapes at the swelled and shrunk states in response to temperature change (Fig. 2a–c, j). The 3D structures at the swelled state adopt new metrics, determined by the areal swelling ratios (Fig. 1c) and the growth functions designed for the target shapes at the shrunk state. Because of the inverse relationship between the areal swelling and shrinking ratios (Fig. 1b, c), 3D shapes with $K > 0$ (e.g., spherical cap in Fig. 2a, right) at the shrunk state in general transform to 3D shapes with $K < 0$ at the swelled state (e.g., saddle-like shape in Fig. 2a, left) and vice versa (Fig. 2b).

In addition to determining $\Omega$ for a target 3D shape (Fig. 2a, c, Supplementary Note 3), we can predict the 3D shape for a given $\Omega$ (Fig. 2l, Supplementary Note 4). To validate the predictive power of the model, we considered a growth function in the form $\Omega(r) = c[1 + (r/R')^2]^{\alpha-1}$, where $R' = aR$ and $a$ and $\alpha$ are constants (Fig. 2l–n, Supplementary Note 4). The theoretical model predicts 3D shapes that consist of a spherical cap-like shape with a smooth gradient in $K$ in the central region and a cone-like shape ($K = 0$) in the outer region (Fig. 2l, Supplementary Note 4). The experimental structures agree well with the theoretical predictions (Fig. 2l–n, Supplementary Figure 6). For example, the base angle $\gamma$ of the experimental structures decreases with $\alpha$, following the predictions (Fig. 2n, Supplementary Note 4). The examples in Fig. 2 illustrate the accuracy and sensitivity of

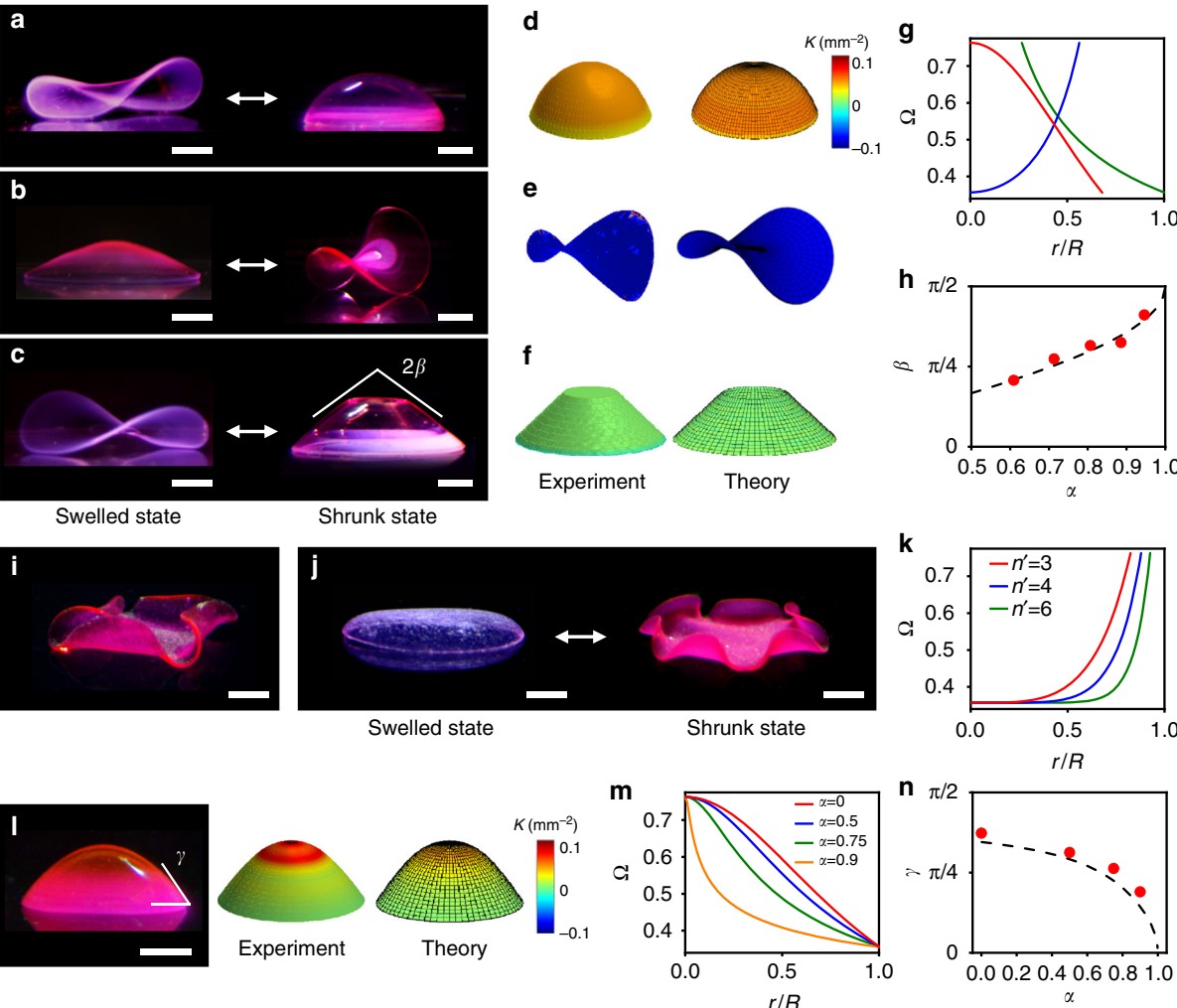

**Fig. 2** Shape-morphing 3D structures with axisymmetric metrics. **a–c** 3D structures with constant Gaussian curvature $K$ at the shrunk state (right) and the corresponding structures at the swelled state (left): spherical cap (**a**), saddle (**b**), and cone (**c**) shapes. **d–f** Reconstructed 3D images with $K$ of experimental (left) and theoretical (right) shapes of the spherical cap (**d**), saddle (**e**), and cone (**f**) structures in **a–c**. **g** $\Omega$ used to form the structures in **a–c**: red line (spherical cap), blue line (saddle), and green line (cone). **h** Experimental (solid circles) and theoretical (dashed line) values of $\beta$ with different $\alpha$ in $\Omega$ for cone structures. **i**, **j** Enneper's minimal surfaces with $n'$ wrinkles: $n' = 4$ (**i**) and 6 (**j**). The structure reversibly transforms between prescribed 3D shapes at the swelled (left) and shrunk (right) states as shown in **j**. **k** $\Omega$ used to form the Enneper's minimal surfaces with $n'$ wrinkles in **i**, **j** and Supplementary Figure 5: $\Omega$ with $n' = 3$ (red line), $n' = 4$ (blue line), and $n' = 6$ (green line). **l** Experimentally constructed 3D structure with a smooth gradient in $K$ (left) and reconstructed 3D images with $K$ of the experimental (middle) and theoretical (right) structures. $\gamma$ is the base angle of the structures. **m** $\Omega$ with different $\alpha$ used to form the structures in **l** and Supplementary Figure 6: $\Omega$ with $\alpha = 0$ (red line), $\alpha = 0.5$ (blue line), $\alpha = 0.75$ (green line), and $\alpha = 0.9$ (orange line) (Supplementary Note 4). **n** Experimental (solid circles) and theoretical (dashed line) values of $\gamma$ of the structures formed with $\Omega$ in **m** as a function of $\alpha$. Scale bars, 5 mm (left), 2 mm (right) in **a–c**; 2 mm in **i**; 5 mm (left), 2 mm (right) in **j**; 2 mm in **l**

our approach. Small changes in $\Omega$ can induce substantial changes in the resulting 3D shapes (e.g., Fig. 2i–k, Supplementary Figure 5)[38].

**Design rules for creating complex 3D structures**. We next sought to establish design rules for creating nonaxisymmetric 3D structures with diverse morphologies (Fig. 3). Our schemes involve the combination and transformation of target metrics and the concept of modularity. As they are implemented in the metric space, these schemes require design rules for how to interface metrics[4]. We thus introduced the concepts of linkers and transitional components at the interfaces of metrics. The radial and azimuthal combinations of growth functions yield hybrid 3D structures with alternating features of the functions along the $r$ (Fig. 3a, b) and $\theta$ (Fig. 3c) directions, respectively, where $\theta$ is an angular position. For example, the radial combination of $\Omega_1$ for a

spherical cap and $\Omega_2$ for a saddle shape induces a structure with $K > 0$ and $K < 0$ in the central and outer regions, respectively (Fig. 3a, Supplementary Figure 7). Another example is a hybrid structure that combines a spherical cap and a cone (Fig. 3b). The azimuthal combination of $\Omega_1$ ($5° < \theta < 85°$ and $185° < \theta < 265°$) and $\Omega_2$ ($95° < \theta < 175°$ and $275° < \theta < 355°$) shown in Fig. 3d yields a structure with alternating features of $\Omega_1$ and $\Omega_2$ along $\theta$ (Fig. 3c). We introduced a linear linker with a form $\Omega_L = (\Omega_1 - \Omega_2)\theta/\Delta\theta + \Omega_2$ with $\Delta\theta = 10°$ at the interfaces of $\Omega_1$ and $\Omega_2$ to make $\Omega$ continuous, as sharp discontinuities in $\Omega$ can cause stress accumulation and thereby shape distortion (e.g., hybrid $\Omega$ without linkers or linkers with $\theta = 5°$, Supplementary Figure 8).

Transforming axisymmetric $\Omega$ into a function of $\theta$ in the form $\Omega(r, \theta) = c(\theta)\Omega(r/(a(\theta)R))$ leads to nonaxisymmetric structures with varying morphologies along $\theta$ (Fig. 3e–l). $c(\theta)$ scales $\Omega$ along $\theta$. Therefore, transforming $\Omega(r) = c(r/R)^2 + \Omega_{min}$ for a modified

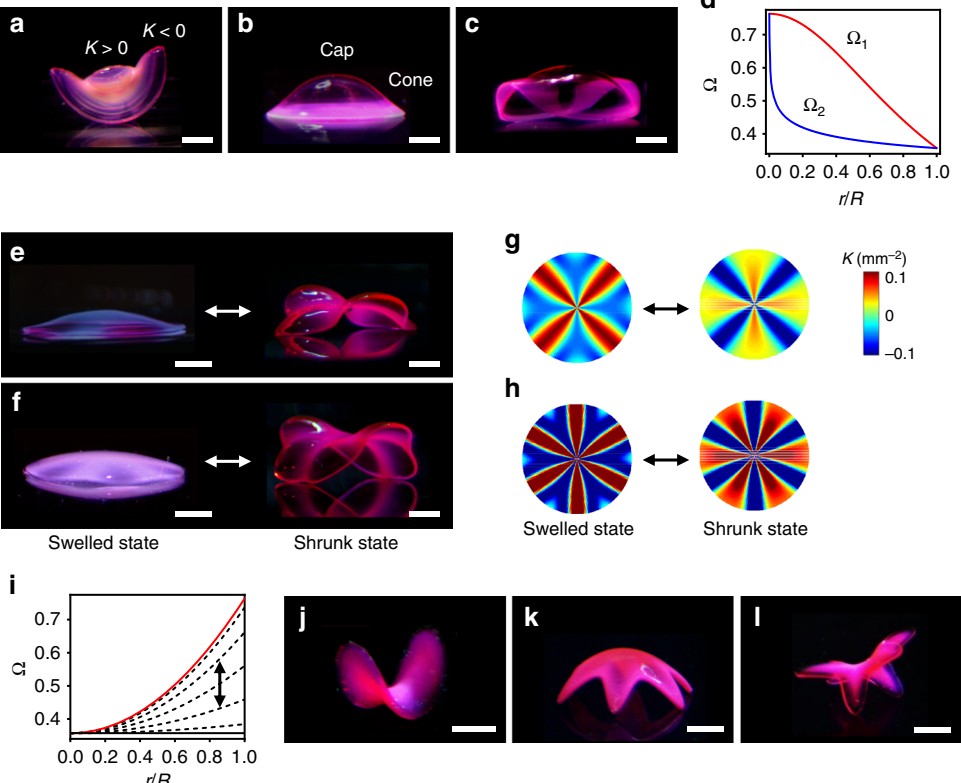

**Fig. 3** Nonaxisymmetric 3D structures with morphological diversity. **a–c** Hybrid 3D structures with radially (**a**, **b**) and azimuthally (**c**) combined $\Omega$. **d** $\Omega$ used to create the 3D structures in **c**: $\Omega_1$ (red line), $\Omega_2$ (blue line). **e**, **f** 3D structures with alternating $K > 0$ and $K < 0$ and 4 (**e**) and 6 (**f**) nodes along $\theta$. **g**, **h** Theoretically calculated Gaussian curvature $K$ of the structures with 4 (**g**) and 6 (**h**) nodes shown in **e**, **f** at the swelled (left) and shrunk (right) states. **i** $\Omega$ used to form the structures in **e**, **f**. The red line, black line, and dashed black lines indicate $\Omega$ at $\theta = 0$ and $(l\pi)/L$ (maximum $\Omega$), $\theta = (2l - 1)\pi/2L$ (minimum $\Omega$), and $\theta$ between the maximum and minimum of $\Omega$, where $l$ and $L$ are constants. $l$ is a positive integer. **j** Elongated elliptical saddle structure with an aspect ratio of 2 ($b = 0.5$). **k** Spherical cap with 6 legs ($b = 0.5$, $L = 3$). **l** Saddle-like structure with 6 legs ($b = 0.5$, $L = 3$). Scale bars, 2 mm in **a–c**; 5 mm (left), 2 mm (right) in **e**, **f**; 2 mm in **j-l**

excess cone (Supplementary Figure 9)[45,46] with $c(\theta) = c_0 \cos^2(L\theta)$, where $\Omega_{min}$ and $L$ are constants, forms a shape with alternating $K > 0$ and $K < 0$ and a programmed number of nodes $n' = 2L$ (Fig. 3e–i, Supplementary Figure 9, Supplementary Movie 2). The resulting structures with 4 ($L = 2$) and 6 ($L = 3$) nodes are shown in Fig. 3e, f, respectively. The structures have the same number of nodes at the swelled and shrunk states (Fig. 3e–h). On the other hand, $a(\theta)$ in $\Omega(r, \theta) = c\Omega(r/(a(\theta)R))$ scales $\Omega$ along $r$. This transformation defines the boundary of structures, while maintaining the functional form and thereby the shape along $r$. Transforming $\Omega$ for a saddle shape with $a(\theta) = \sqrt{1 + (b^2 - 1)\sin^2\theta}$ thus forms an elongated elliptical saddle structure with an aspect ratio of $1/b$ ($0 < b < 1$) (Fig. 3j, Supplementary Figure 10); an elongated saddle structure with an aspect ratio of 2 ($b = 0.5$) is shown in Fig. 3j. Interestingly, the directions of the principal curvatures at the center of the saddle structure align with the major and minor axes of the ellipse, suggesting that this configuration is an embedding of the lowest bending energy of the target metric. Furthermore, adding periodicity into $\Omega$ with $a(\theta) = \sqrt{1 + (b^2 - 1)\sin^2(L\theta)}$ modulates the number of nodes $n' = 2L$ along $\theta$. Using this transformation, we could form spherical cap and saddle-like structures with a targeted number of legs (Fig. 3k, l, Supplementary Figure 11). The examples in Fig. 3 show the versatility of our approach in creating diverse 3D morphologies. These structures can be further used as a building block for multimodular 3D structures (Fig. 4).

**Multimodular 3D structures.** The modular assembly of target metrics can create 3D structures with broad morphological and functional diversity (Fig. 4, Supplementary Figure 12)[4]. However, there is an intrinsic problem in assembling modules in the metric space[4]. Each module can randomly adopt the direction of deformation (e.g., upward or downward)[4] or the orientation with respect to other modules due to the symmetric nature of metrics. Thus, a multimodular $\Omega$ can in general form multiple different conformations presumably with the same elastic energy (as shown in previous work[4] and Supplementary Figure 12). To tackle this problem, we introduced the concept of transitional components, designed to control the direction of deformation and the orientation of modular components (Fig. 4a–j). A saddle-like structure with $K < 0$ (e.g., Figs. 2b, 3j) has the principle curvatures with the same sign along its parallel edges. We thus postulated that modular components with $K > 0$ that share the parallel edges of a saddle-like structure ($K < 0$), or a parallel transitional component (e.g., small circles with dashed white lines in Fig. 4f–i), would deform in the same direction as the parallel edges. On the other hand, modular components with $K > 0$ that share the perpendicular edges of a saddle-like structure ($K < 0$), or a perpendicular transitional component (e.g., large circles with dashed white lines in Fig. 4f–i), would deform in the opposite directions. Figure 4a–d shows examples of multimodular structures with directional control, where the arrows indicate the programmed orientation of each module. Placing the parallel and perpendicular transitional components between modules

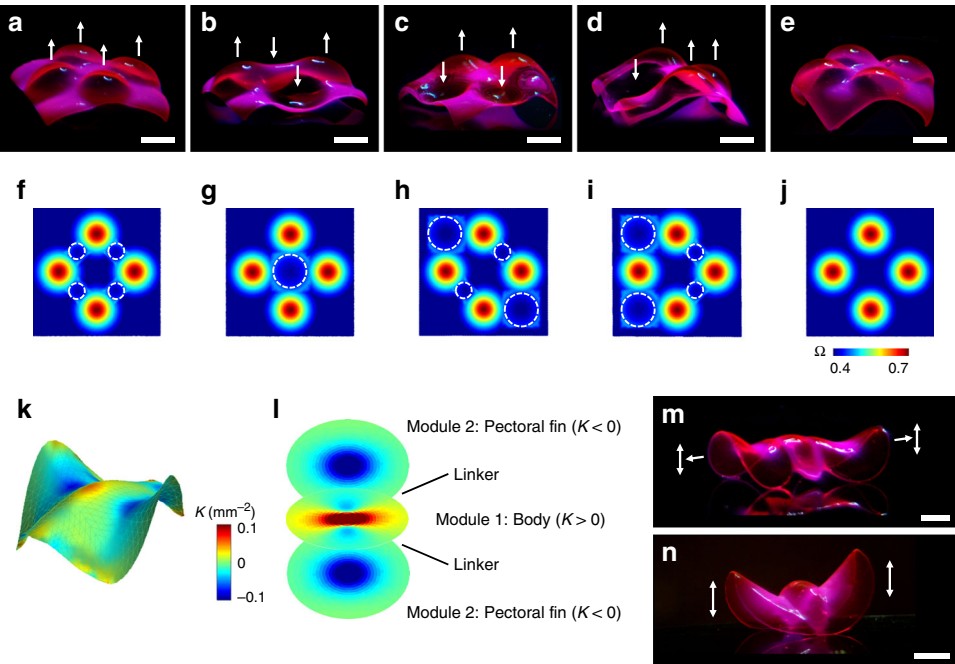

**Fig. 4** Multimodular 3D structures. **a–e** Examples of multimodular 3D structures with 4 modules with (**a–d**) and without (**e**) directional control. The modules were programmed to deform in the directions indicated by the white arrows. **f–j** Strategies to control the orientation of the modules in the corresponding structures in **a–e**. The color maps illustrate $\Omega$ used to create the structures. The small and large circles with white dashed lines indicate the parallel and perpendicular transitional components, respectively. **k** Reconstructed 3D image of a stingray model with $K$. **l** Modular design of a stingray-inspired 3D structure in **m**. The modules for the body and the pectoral fins were designed based on the $K$ map in **k** and Supplementary Figure 13. **m, n** Stingray-inspired 3D structures with oscillatory flapping motions. The white arrows indicate the direction of the motions. Scale bars, 4 mm in **a–e**; 2 mm in **m**; 4 mm in **n**

(indicated by the small and large circles with white dashed lines in Fig. 4f–i, respectively) led to the multimodular structures with designed morphologies (Fig. 4a–d). In contrast, the modules in a control structure without transitional components tend to deform in the same direction, implying slight variations in shrinkage through the thickness; the variations make a specific direction energetically favorable for all modules (Fig. 4e, j).

The design rules established in this work (Figs. 3, 4) offer simple yet versatile ways to build complex 3D structures without the need for extensive computation. To demonstrate this capability, we fabricated ray-inspired 3D structures that replicate the key morphological features of stingrays, including the pectoral fins with $K < 0$ (Fig. 4k–n)[21,47]. We designed multimodular structures based on the reconstructed 3D images, $K$, and swimming motions of stingrays (Fig. 4k, l, Supplementary Figure 13)[21,47]. The growth functions for the body and the pectoral fins were designed and merged with linear linkers (Fig. 4l), using the design rules shown in Figs. 3, 4a–j (Supplementary Note 5). For example, the module for the body structure with the linkers was used as a transitional component that controls the orientation of the left and right pectoral fins with respect to the body and thus synchronizes their motions (Supplementary Figure 14). Furthermore, the ray-inspired structures were designed to produce different types of oscillatory flapping motions in response to temperature cycles (between 31.5 °C and 33.5 °C), mimicking those of stingrays (Supplementary Movies 3, 4).

**Dynamic behavior of growth-induced 3D structures.** We next explored how growth-induced 3D structures transform their shapes (Fig. 5). To elucidate the underlying mechanism of the shape evolution, we introduced the concept of dynamic target metrics (Fig. 5a–e). To verify this concept, we used a spherical cap structure shown in Fig. 2a as our model system. Figure 5a shows the shape evolution of a spherical cap. Despite its simple shapes at equilibrium (swelled and shrunk states), the structure undergoes complex shape transformations. Our results reveal that the spatially nonuniform rates of swelling and shrinking of growth-induced 3D structures determine their dynamic shape changes as described below.

To understand the dynamic behavior, we first measured $A_T/A_0$ of homogeneous hydrogels (i.e., hydrogel disks uniformly cross-linked by $t_{ex}$) as a function of time $t$ during cooling (Fig. 5b). The measurements indicate that the swelling rates decrease with $t_{ex}$ (as shown in Fig. 1d), reflecting the difference in the rate of diffusion of water through the hydrogels with different densities (supporting the mechanism in Fig. 1). The crossover of $A_T/A_0$ of hydrogels prepared with short and long $t_{ex}$ at around 30–35 min (indicated by the dashed black line in Fig. 5b) implies how growth-induced structures transform between shapes with $K > 0$ and $K < 0$ (e.g., Figs. 2a–c, j, 5a).

To quantitatively describe the shape evolution, we next constructed dynamic calibration curves ($A_T/A_0$ as a function of $t_{ex}$ at times $t$) using $A_T/A_0$ shown in Fig. 5b (Fig. 5c, Supplementary Figure 15), analogous to the static calibration curves (Fig. 1b, c). The dynamic calibration curves show how the local areas created with $t_{ex}$ in 3D structures swell (or shrink) with $t$. $A_T/A_0(t_{ex})$ changes from the static calibration curve at the shrunk state (i.e., $A_T/A_0(t_{ex})$ at $t = 0$ min in Fig. 5c) to the static calibration curve at the swelled state (Supplementary Figure 15). We can then determine how $\Omega$ for a 3D shape evolves with $t$ (dynamic growth function or target metric $\Omega_t$) from $\Omega$ at $t = 0$ min (i.e., $\Omega$ at the shrunk state), using the dynamic calibration curve $A_T/A_0(t_{ex})$ at $t$.

Having established the procedure to determine $\Omega_t$, we applied the concept of dynamic target metrics to investigate the shape evolution of the spherical cap structure (Fig. 5a). We determined

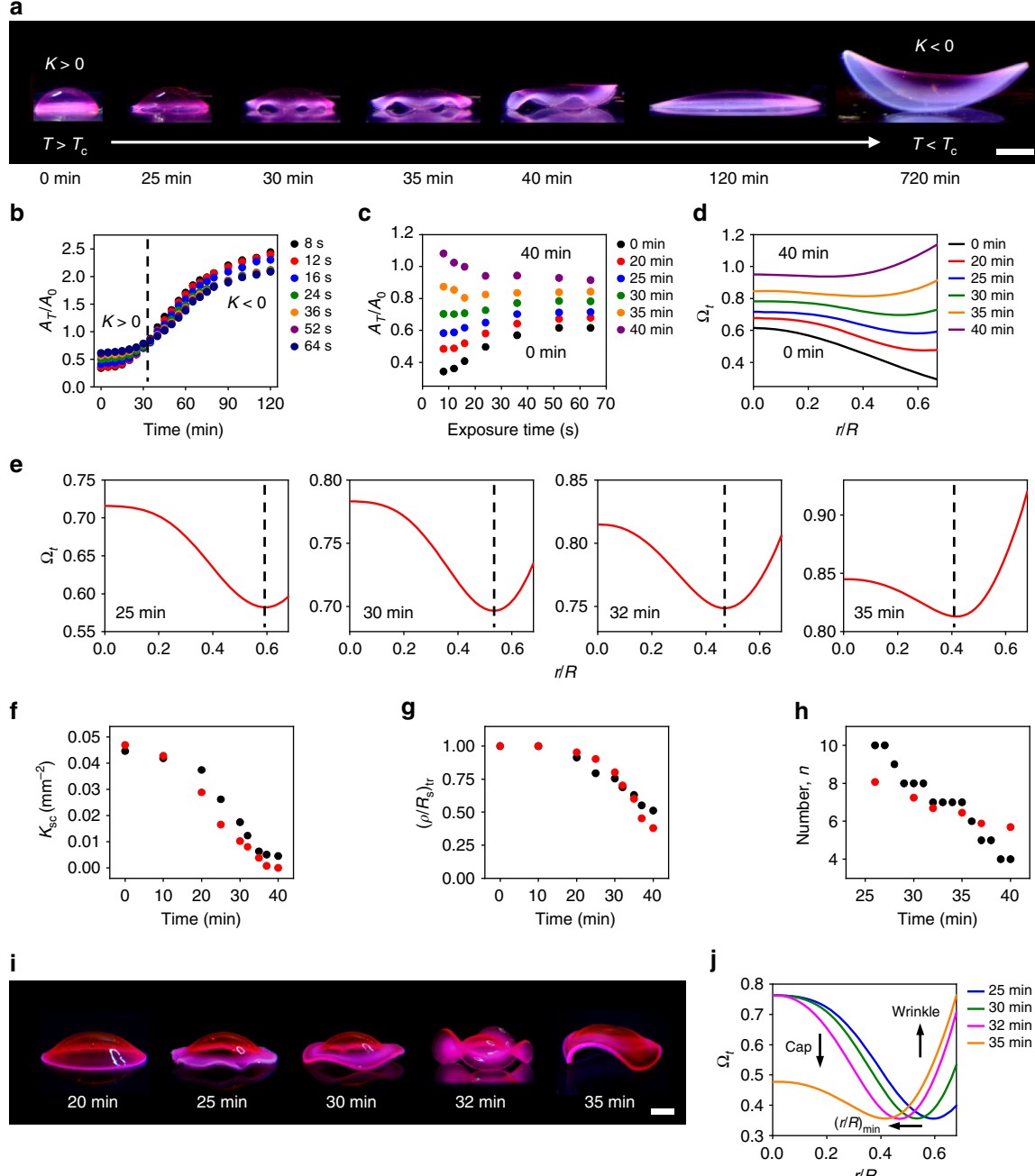

**Fig. 5** Dynamic behavior of growth-induced 3D structures. **a** Shape evolution of a spherical cap during cooling. **b** $A_T/A_0$ of homogeneous hydrogels formed with different $t_{ex}$ as a function of cooling time $t$. The black, red, blue, green, orange, purple, and navy circles represent $A_T/A_0$ of the hydrogels formed with $t_{ex}$ of 8, 12, 16, 24, 36, 52 and 64 s, respectively. **c** $A_T/A_0$ in **b** as a function of $t_{ex}$ at different $t$ (dynamic calibration curves). The black, red, blue, green, orange, and purple circles represent $A_T/A_0$ at $t$ of 0, 20, 25, 30, 35 and 40 min, respectively. **d** Dynamic growth function (or target metric) $\Omega_t$ of the spherical cap at $t = 0$ to 40 min. The black, red, blue, green, orange, and purple lines represent $\Omega_t$ at $t$ of 0, 20, 25, 30, 35 and 40 min, respectively. **e** $\Omega_t$ for the shapes of the spherical cap in **a** at 25, 30, 32 and 35 min. **f** Experimentally measured $K_{sc}$ ($K$ of the spherical cap-like shape in the center) (black circles) and theoretically calculated $K_{sc}$ (red circles) as a function of $t$. **g** Experimentally measured $(\rho/R_s)_{tr}$ (location of the shape transition) (black circles) and theoretically calculated $(\rho/R_s)_{tr}$ (red circles) as a function of $t$. **h** Experimentally measured number of the wrinkles (black circles) as a function of time. The red circles represent $n$ obtained from the fit of $\Omega = c/[1 + (r/(aR))^2]^2 + [1 + (r/R)^n]^2 - 1$ to $\Omega_t$ as described in the main text and Supplementary Figure 20. **i** Replicated structures of the dynamic shapes of the spherical cap in **a** at 20, 25, 30, 32 and 35 min. **j** Normalized $\Omega_t$ used to create the structures shown in **i**. The blue, green, magenta, and orange lines represent $\Omega_t$ at 25, 30, 32 and 35 min, respectively. Scale bars, 5 mm in **a**; 2 mm in **i**

$\Omega_t$ for the spherical cap (Fig. 5d, e, Supplementary Figure 16) from its static growth function (shown in Fig. 2g), using the dynamic calibration curves (shown in Fig. 5c, Supplementary Figure 15). $\Omega_t$ shows how the metric of the spherical cap changes with time and thus how the structure changes its shape (Fig. 5d, e,

Supplementary Figures 16, 17). During this transition, $\Omega_t$ undergoes complex transformations, forming hybrid elliptic and hyperbolic metrics (Fig. 5d, e) and thus inducing hybrid 3D shapes (Fig. 5a, Supplementary Figure 18). The spatially nonuniform kinetics of swelling produces hybrid $\Omega_t$ with a minimum at

$(r/R)_{min}$ at $t$ of 20–40 min (as indicated by dashed black lines in Fig. 5e and Supplementary Figure 18). $\Omega_t$ at $r/R < (r/R)_{min}$ and $r/R > (r/R)_{min}$ represents the spherical cap-like shape in the center ($K > 0$) and the wrinkles in the edge ($K < 0$), respectively. The functional form of $\Omega_t$ (e.g., sharp change in the gradient of $\Omega_t$ at $r/R > (r/R)_{min}$) reflects how $\Omega_t$ forms wrinkles, reminiscent of Enneper's surfaces (Fig. 2i–k, Supplementary Figure 5).

To demonstrate that $\Omega_t$ predicts the dynamic shape change, we quantified the shape evolution of the spherical cap structure at 0–40 min and compared it with our theoretical model (Fig. 5f–h, Supplementary Figures 18–20). We characterized the dynamic shapes by Gaussian curvature of the spherical cap-like shape in the center $K_{sc}$ (Fig. 5f), the location of the shape transition between the spherical cap-like shape ($K > 0$) and the wrinkles ($K < 0$) $(\rho/R_s)_{tr}$, where $\rho$ is the radial coordinate of the 3D structure and $R_s$ is the radius of the structure (Fig. 5g, Supplementary Figures 18, 19), and the number and amplitude of wrinkles (Fig. 5h, Supplementary Figure 20). The experimentally measured $K_{sc}$ decreases with time and the spherical cap-like shape gradually disappears at around 40 min ($K \rightarrow 0$), matching well with $K_{sc}$ obtained from $\Omega_t$ (Fig. 5f). The measured $(\rho/R_s)_{tr}$ decreases with time (i.e., shifts toward the center of the structure), showing a good agreement with $(\rho/R_s)_{tr}$ calculated from $(r/R)_{min}$ (using Supplementary Equation 8 in Supplementary Note 2). The shift of $(\rho/R_s)_{tr}$ (or $(r/R)_{min}$) results in the decrease in the region of the spherical cap-like shape and the increase in the region of the wrinkles.

Furthermore, our theoretical model describes how the number of the wrinkles decreases with time while their amplitude increases (as shown in Fig. 5a and Supplementary Figure 18). To understand how the structure forms the wrinkles, we fitted $\Omega = c/[1 + (r/(aR))^2]^2 + [1 + (r/R)^n]^2 - 1$ to $\Omega_t$, where the first and second terms in $\Omega$ represent the spherical cap-like shape (as shown in Fig. 2g) and the wrinkles (a functional form of Enneper's surfaces in Fig. 2k), respectively, and $c$, $a$, and $n$ are constants. The results show that $n$ decreases with time, suggesting that the decrease in $n$ results in the decrease in the number of the wrinkles and the increase in their amplitude (Fig. 5h, Supplementary Figure 20), as observed in Enneper's surfaces with different $n'$ (Fig. 2i–k, Supplementary Figure 5)[4,37]. The dynamic $K$ maps theoretically calculated from $\Omega_t$ reflect the experimentally observed shape transformations (Supplementary Figure 21).

To further demonstrate that $\Omega_t$ can predict the dynamic behavior of growth-induced 3D structures, we replicated the dynamic shapes of the spherical cap using $\Omega_t$ (Fig. 5i). Because the full range of $\Omega_t$ is not accessible by our material systems, we rescaled $\Omega_t$ to the experimentally accessible range of $\Omega$ (Fig. 5j). The replicated structures reproduce the key signatures of the shape evolution, including the formation of wrinkles and their shape changes (e.g., increase in the amplitude of wrinkles, decrease in their number, and gradual disappearance of the spherical cap-like shape in the center) (Fig. 5i). The discrepancy in the detailed shapes (e.g., enhanced wrinkles) is attributed to the use of normalized $\Omega_t$. Moreover, this approach that uses $\Omega_t$ for 3D shaping provides new pathways for creating complex 3D structures. This approach offers rich sources to design complex 3D shapes, difficult to access with current theories (e.g., wrinkle formation)[35], and to understand how differential in-plane growth translates to 3D shapes. Manufacturing complex 3D structures, such as those shown in Fig. 5i, is difficult and expensive to achieve by other methods.

**Dynamic 3D structures with programmed sequential motions.** Another important finding is that the swelling and shrinking rates of our hydrogel systems are phototunable and thus locally

programmable. To demonstrate the ability to control the speed of shape change, we created saddle structures with an identical shape but different speeds of shape transformation (Fig. 6a, b, Supplementary Figure 22). To create these structures, we designed $\Omega_{fast}$ and $\Omega_{slow}$ with the same functional form ($\Omega$ for a saddle shape shown in Fig. 2g) but in different $\Omega$ ranges: $\Omega_{slow}/\Omega_{fast} = C$, where $C$ is a constant and $C > 1$ (Fig. 6c). The growth (swelling and shrinking) rates decrease with $t_{ex}$ and thus with $\Omega$ (Fig. 1d, Supplementary Figure 23), but a 3D shape is determined by the relative growth (not by the absolute values of $\Omega$). We can therefore program the speed of shape transformation without changing 3D shapes by controlling the range of $\Omega$ (e.g., the maximum and minimum values of $\Omega$) but maintaining the relative growth (e.g., $\Omega(r/R)/\Omega_{min}$, where $\Omega_{min}$ is a constant). As designed, the structure with $\Omega_{fast}$ transforms its shape faster than the structure with $\Omega_{slow}$ (Fig. 6a, b, Supplementary Figure 22). The dynamic $K$ maps theoretically calculated from the dynamic growth functions describe the experimentally observed shape transformations with different speeds (Fig. 6b, Supplementary Figure 22). Within the structures (Fig. 6a, b), due to the difference in the range of $\Omega$, the central regions ($r/R \sim 0$; low range $\Omega$) transform faster than the edge regions ($r/R \sim 0.4$; high range $\Omega$), also seen in the dynamic $K$ maps (Fig. 6b, Supplementary Figure 22). The same trend is observed in the spherical cap structure (Fig. 5a), in which the edge region (low range $\Omega$) transforms faster than the central region (high range $\Omega$) (Supplementary Figure 21).

The ability to spatially control the rate of shape transformation allows us to create dynamic 3D structures with programmed sequential motions, difficult to achieve with global external stimuli[8]. As a demonstration, we fabricated a ray-inspired 3D structure with programmed sequential motions (Fig. 6d, e, Supplementary Figures 24, 25). The structure consists of modules for the body ($K > 0$), front wings ($K < 0$), and rear wings ($K < 0$) (Supplementary Figure 24). The front and rear wings were designed to transform fast and slowly, respectively, and thereby be sequentially actuated in response to temperature change (Fig. 6d, e). As designed, the front wings transform first from a shape with $K_c < 0$ to $K_c > 0$ (around 5 min), gradually lifting the rear wings, while the rear wings slowly transform (e.g., $K_c < 0$ up to 10 min) and flap after 20 min (Fig. 6d, Supplementary Movie 5). Moreover, we can control the oscillatory motions (e.g., amplitude and frequency) by modulating temperature cycles (Supplementary Figure 25). The theoretically calculated dynamic $K$ maps for each module illustrate the experimentally observed sequential motions (Fig. 6e).

## Discussion

Living organisms often achieve 3D morphologies and movements by using spatially patterned and temporally controlled expansion and contraction of continuously deformable soft tissues. Our approach that uses the spatially and temporally controlled growth for programming 3D shapes and their motions, possibly with a large number of degrees of freedom, could thus create dynamic 3D structures that mimic the morphologies and motions of living organisms and thus, potentially, their functions. The ability to program growth-induced 3D shapes and motions could potentially transform the way we design and fabricate soft engineering systems, such as soft robots, actuators, and artificial muscles. The concept is applicable to other programmable materials. The 2D printing approach for 3D material programming represents a scalable and customizable 3D manufacturing technology, potentially integrable with biological systems[1,2,7,21,30] and existing 2D fabrication methods and devices for multifunctionalities and broader applications[48].

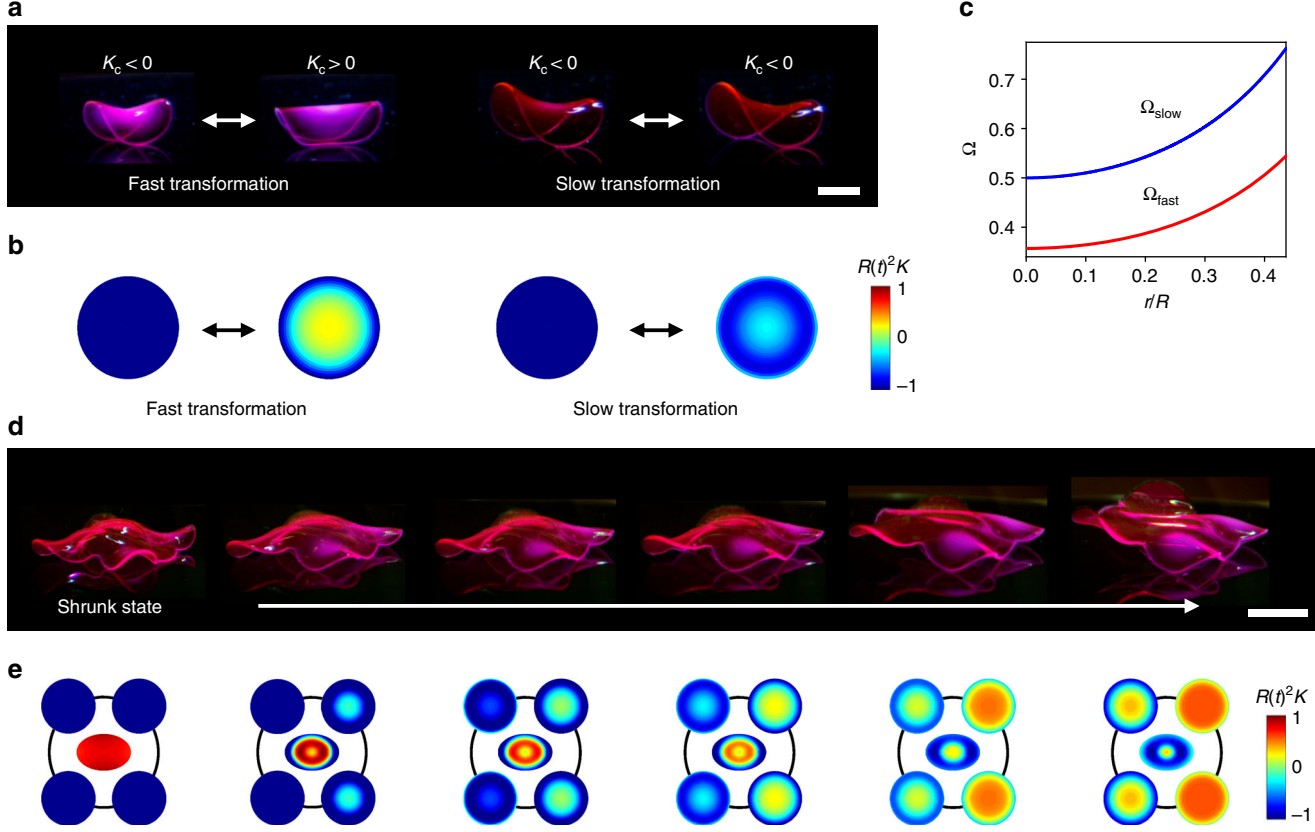

**Fig. 6** Dynamic 3D structures with programmed sequential motions. **a** Fast and slowly transforming saddle structures at $t = 0$ (left) and 30 min (right) during cooling. $K_c$ is $K$ in the center of structures. **b** Theoretically calculated $K$ of the structures in **a**. $R(t)^2 K$ is normalized $K$ using the time-dependent $R$ as described in Supplementary Figure 21. **c** Growth functions used to create the saddle structures with different speeds of shape transformation in **a**. The red and blue lines represent $\Omega_{fast}$ and $\Omega_{slow}$ used to create the fast and slowly transforming structures in **a**, respectively. **d** Ray-inspired 3D structure with programmed sequential motions. The images show the structure at the equilibrium shrunk state and the dynamic structures at 0, 5, 10, 20 and 25 min during cooling of a temperature cycle (from left to right). **e** Theoretically calculated dynamic $K$ maps of the modules in the structures in **d** at the equilibrium shrunk state and at 0, 5, 10, 20 and 25 min during cooling (from left to right). Scale bars, 2 mm in **a**; 5 mm in **d**

## Methods

**Preparation of precursor solutions.** The precursor solutions for pNIPAm crosslinked with BIS and PEGDA were prepared by dissolving NIPAm (0.4 g), BIS (0.5 mol% of NIPAm), PEGDA with an average molecular weight (MW) of ~700 g mol⁻¹ (0.125 mol% of NIPAm), and diphenyl(2,4,6-trimethylbenzoyl)phosphine oxide (PBPO) (0.15 mol% of NIPAm) in 1 mL aqueous solutions (1:3 ratio of water and acetone by volume). The precursor solutions for pNIPAm crosslinked with BIS were prepared by dissolving NIPAm (0.2 g), BIS (0.25–5.0 mol% of NIPAm), and PBPO (0.3 mol% of NIPAm) in 1 mL aqueous solutions (1:3 ratio of water and acetone). The precursor solutions for pNIPAm crosslinked with PEGDA were prepared by dissolving NIPAm (0.2 g), PEGDA (1.0–10.0 mol% of NIPAm), and PBPO (0.3 mol% of NIPAm) in 1 mL aqueous solutions (1:3 ratio of water and acetone). All materials were purchased from Sigma-Aldrich and used as received.

**Creation of shape-morphing 3D structures.** Projection lithography cells were prepared by placing a polydimethylsiloxane (PDMS) spacer with a thickness of 400 μm on a PDMS substrate. After purging with nitrogen to reduce the effects of oxygen on photopolymerization, the precursor solutions were introduced into the cells. The cells were then covered with a glass coverslip (150 μm in thickness). The precursor solutions were then programmed with growth functions (or target metrics) Ω by spatially and temporally controlled ultraviolet (UV) light (dynamic light projection grayscale lithography).

Shape-morphing 3D structures were created using DL4P. 2D structures that define the boundary of target structures were designed using 3ds Max (Autodesk). Growth functions Ω designed for target 3D shapes were converted into 2D maps of light exposure times using the calibration curves of the areal swelling and shrinking ratios versus light exposure time (Fig. 1b, c) with a custom-made MATLAB (MathWorks) code. The growth functions define local $A_T/A_0$ of 2D structures. Stereolithography (STL) files were generated that contain the information of the 2D maps of light exposure times. The precursor solutions were polymerized and crosslinked by spatially and temporally controlled UV light using a digital light processing (DLP) projector (Vivitek D912HD) with the STL files (dynamic light projection lithography). After polymerization and crosslinking, the 2D hydrogel

structures were detached from the cell and immediately washed with acetone, isopropyl alcohol (IPA), and water for three times to remove unreacted monomers, crosslinkers, and photoinitiators, and suppress photopolymerization and crosslinking reactions. To achieve the target 3D shapes at the equilibrium swelled state, the hydrogel structures were immersed in water at 4 °C for 72 h, while exchanging the water every 12 h, and then at 25 °C for 2 h in a temperature controlled water bath. To induce the target 3D shapes at the equilibrium shrunk state, the temperature of the water was slowly increased to 35 °C. Food color dyes were introduced into hydrogel structures for imaging. Hydrogel structures without dyes are transparent at equilibrium states.

**Measurement of areal swelling and shrinking ratios.** Homogeneous pNIPAm hydrogel disks with a diameter of 5 mm (i.e., hydrogel disks formed with constant Ω) were prepared by DL4P. The hydrogel disks were uniformly exposed to UV light over the entire disks with light exposure times of 8–70 s. The areas of the hydrogel disks at the swelled state $A_{25}$ were measured at 25 °C. The areas of the hydrogel disks at the shrunk state $A_{35}$ were measured at 35 °C. The areal swelling and shrinking ratios are defined as $A_{35}/A_0$ and $A_{25}/A_0$, respectively, where $A_0$ is the area of as-prepared hydrogel disks. The hydrogel disks were used to generate the calibration curves of the areal swelling and shrinking ratios versus light exposure time (Fig. 1b, c, Supplementary Figure 1). This process induces essentially no or little variation of swelling and shrinking through the thickness and thus does not induce bending of homogeneous hydrogel disks.

**Measurement of mechanical properties and gel points.** The dynamic mechanical properties of hydrogels at the swelled state were measured using a rheometer (DHR-2, TA Instruments) with a 20-mm plate geometry. Hydrogel disks with a diameter of 20 mm were used. The shear storage modulus $G'$ and loss modulus $G''$ were measured by frequency sweeps of 0.01–100 rad s⁻¹ at an oscillatory strain of 1%. The hydrogel disks with a storage shear modulus larger than 20 Pa were used for the measurements of the swelling and shrinking ratios (Fig. 1b, c, Supplementary Figure 1).

The gel points of hydrogels crosslinked with single crosslinkers were measured by the method of Winter and Chambon (Supplementary Figure 2)[49,50]. Hydrogel disks with a diameter of 20 mm were prepared with single crosslinkers (BIS and PEGDA; 1 mol% of NIPAm in precursor solutions) with different light exposure times (BIS-crosslinked hydrogels: 4, 8, 12 and 16 s; PEGDA-crosslinked hydrogels: 1, 2, 3 and 4 s). The hydrogel disks that form stable hydrogels after washing with acetone and IPA were used for the measurements. $G'$ and $G''$ were measured by frequency sweeps of 0.1–15 Hz at an oscillatory strain of 0.1% using the rheometer with a 20-mm plate geometry. To determine the gel points, $\tan\delta = G''/G'$ were plotted as a function of frequency. At the gel point, $\tan\delta = G''/G'$ has a constant value over the frequency sweep (Supplementary Figure 2)[49,50].

**Reconstruction of 3D images and Gaussian curvatures**. The 3D images and Gaussian curvatures $K$ of experimentally created 3D structures were constructed based on the spin image 3D recognition method[51]. 2D images of the 3D structures were captured from different angles by taking images while rotating the structures. 3D images were then reconstructed from the 2D images and converted into STL files using 3ds Max. The reconstructed 3D images with $K$ were constructed using MATLAB with the STL files (Fig. 2d–f, l, Supplementary Figure 6).

## Data availability
The data that support the findings of this study are available within the paper and its Supplementary Information and from the corresponding author upon reasonable request.

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

## Acknowledgements
We thank C. Kim, S. Koh, H. F. Tibbals, A. Aghamohammadi, and M. Khorrami for discussion.

## Author contributions
A.N. and K.Y. conceived the project and designed the experiments. A.N. and H.A. performed the experiments. A.N, H.A., K.L. and K.Y. analyzed the data. A.N. and K.Y.

wrote the manuscript. All authors discussed data and contributed to the writing of the manuscript.

## Additional information

**Competing interests:** The authors declare no competing interests.

