## [Peer Review File · Nature Communications]

Reviewers' comments:

Reviewer #1 (Remarks to the Author):

This paper simultaneously makes bold claims of novelty that, I think, are not entirely fair but it also introduces some nice - actually, very nice - results.

For example, the paper is written to imply that they are, perhaps, the first to produce complex shapes similar to those of living things but they don't demonstrate shapes that are any more complex than those in Ref. 4-5 (and probably others). Those in Ref. 5 are particularly living-like by design. In the case of modularity, Ref. 4 already demonstrated mixed metrics in a similar manner to what was demonstrated here. I'm not saying that there isn't something of value here, only that it sounded like the claims to novelty were stronger than the reality.

On the other hand, there are some very nice and very novel aspects. The authors are absolutely correct that the dynamics of non-Euclidean plates remains fairly unexplored and this paper, therefore, makes an important early contribution in understanding those shape dynamics. Unfortunately, I would have liked to see more details about these dynamics and perhaps some effort at quantifying what is happening in some simple and understandable cases. Finally, however, what the authors do with programming shape dynamics at the end of the paper is truly awe-inspiring. I think this is the most important part of the paper and the greatest advance. I would prefer to see more details about how this works as I found the dynamic programming methods rather confusing as presented.

In the end, I like the paper and I think it should be published. But I would prefer there to be less repetition of what was already done with programmed shape and more detail on the dynamical programming and shape dynamics in general. The basic shape programming can still go in the supplement as it is important to verify that everything works as expected and reproduce existing results.

Some nitpicks:

In line 33, growth and shrinkage does not define a shape uniquely but only defines the Gaussian curvature uniquely. There are, generically, many shapes with the same Gaussian curvature for topological disks. I think the authors know this and meant something else.

Line 46-47, I do not think it is fair to the literature to say that mostly axisymmetric shapes are understood. There are several experimental systems with non-axisymmetric shapes and metrics available and, though there are things still to understand, this sentence seems too strong.

Line 157, there is a typo in the word "shape"

Reviewer #2 (Remarks to the Author):

The authors describe some interesting curving and folding results with PEGDA and NIPAM based gels. PEGDA and NIPAM-BIS gels. The tunability of the network is nice and the results are interesting. However, the authors need to more clearly incorporate prior work and correctly reference prior art.

For example, bending and curving by phototunability during crosslinking in a polymer has been discussed previously to create a variety of shapes. (Jamal et al, Differentially photo-crosslinked polymers enable self-assembling microfluidics. Nature communications, 2, p.527, 2011).

Then tuning curvature of folding using PEGDA gels of different molecular weight has also been utilized previously. (Jamal et al, Bio-Origami Hydrogel Scaffolds Composed of Photocrosslinked PEG Bilayers." Advanced healthcare materials 2, no. 8 (2013): 1142-1150)

Finally stating that programmability of shape by diffusion and extent of cross-linking has not been previously achieved is a bit exaggerated. Please see: Cangialosi, A., et al, 2017. DNA sequence-directed shape change of photopatterned hydrogels via high-degree swelling. Science, 357(6356), pp.1126-1130.

Finally, biological organisms do not utilize light to program their shape dynamically, so the link to that is a bit tenuous.

I recommend that the authors tone down some of the text, include references to prior work accurately and certainly the paper can be interesting from a mechanics perspective.

Reviewer #3 (Remarks to the Author):

This paper is difficult to read because it is too full of hype.

From what can be understood from the figures (and repeated re-reading), the authors control the photo-polymerization of the gel

and then once the gel is fabricated, use temperature to control the swelling of the sample. This is a nice notion, but should be explicitly stated as the work. Rather, there is this confusing notion of "growth". They say the word "growth" means expansion and contraction, yet there is not a consistent use of the word in the paper. Does growth also apply to the polymerization phase?

The work is interesting, though it is not clear that it belongs in Nat. Comm. It may be better for a more specialized journal.

Even if it goes to a different journal, the authors should think about rewriting the introduction and more clearly stating what exactly they have done--as I understand it, it is sufficiently interesting that they program the photo-polymerization to control the 3D shape of thermo-responsive gels. In this context, it is worth citing work by Jeremiah Johnson at MIT and K. Matyjaszewski at CMU and then discussing how the new work is different from the latter studies involving photo-polymerization.

Point-by-point Response to the Reviewers' Comments

We thank the reviewers for their constructive and thoughtful comments. Their comments and suggestions greatly helped us to improve our manuscript. According to the reviewers' comments, we revised our manuscript as described below.

Reviewer #1 (Remarks to the Author):

Comment 1. This paper simultaneously makes bold claims of novelty that, I think, are not entirely fair but it also introduces some nice - actually, very nice - results.

Response: We thank the reviewer for the thoughtful and constructive comments. According to the reviewer's comments, we revised our manuscript as described below.

Comment 2. For example, the paper is written to imply that they are, perhaps, the first to produce complex shapes similar to those of living things but they don't demonstrate shapes that are any more complex than those in Ref. 4-5 (and probably others). Those in Ref. 5 are particularly living-like by design. In the case of modularity, Ref. 4 already demonstrated mixed metrics in a similar manner to what was demonstrated here. I'm not saying that there isn't something of value here, only that it sounded like the claims to novelty were stronger than the reality.

Response: We thank the reviewer for the comment. We agree with the reviewer that Ref. 4 (Kim et al., *Science*, 2012) and Ref. 5 (Gladman et al., *Nat. Mater.*, 2016) demonstrated various 3D structures, including those that mimic 3D morphologies of living organisms. These studies (Ref. 4 and 5) are the key references of our study that inspired us to create bioinspired 3D structures. According to the reviewer's comment, we revised our manuscript as described below.

(1) To clearly show that the previous work demonstrated complex 3D shapes, including those with biological morphologies, we rewrote the sentence and added Ref. 4 and 5 in Lines 27–29: “These approaches have been used to build various self-shaping 3D structures, including those with complex 3D morphologies of living organisms^{4,5}, but reproducing their movements has not been fully achieved^{1,2,20}.”

We added Ref. 4 and 5 in Line 34–36: “This approach defines 3D shapes with Gaussian curvatures^{21,22} and is uniquely capable of creating 3D structures with curved geometries, often seen in biological organisms but difficult to achieve by other methods^{4,5}.”

We added Ref. 4 in Line 39–40: “With the physical properties of hydrogels similar to those of soft tissues^{7,12,32}, this approach thus has great potential for creating bioinspired 3D structures⁴.” We did not include Ref. 5 here, as this work used extrusion-based 3D printing of bilayer structures with anisotropic swelling.

(2) To clearly show that Ref. 4 demonstrated mixed metrics, we changed the wording, added a phrase (“as shown in previous work⁴” in Line 222–228), and included Ref. 4 in the following sections.

We added Ref. 4 in Line 64–66: “we established simple yet versatile design rules and the concept of modularity for creating complex 3D structures with diverse morphologies,⁴”

We deleted the word “unexplored” and added Ref. 4 in Line 184–186: “As they are implemented in the metric space, these schemes require design rules for how to interface metrics⁴.”

We added “as shown in previous work⁴” and Ref. 4 and deleted the word “uninvestigated” in Line 221–227: “The modular assembly of target metrics can create 3D structures with broad morphological and functional diversity (Fig. 4, Supplementary Fig. 12)⁴. However, there is an intrinsic problem in assembling modules in the metric space⁴. Each module can randomly adopt the direction of deformation (e.g., upward or downward)⁴ or the orientation with respect to other modules due to the symmetric nature of metrics. Thus, a multimodular Ω can in general form multiple different conformations presumably with the same elastic energy (as shown in previous work⁴ and Supplementary Fig. 12).”

(3) We toned down the text throughout the revised manuscript (especially in the introduction and conclusion). We highlighted the changes in red in the manuscript.

Comment 3. On the other hand, there are some very nice and very novel aspects. The authors are absolutely correct that the dynamics of non-Euclidean plates remains fairly unexplored and this paper, therefore, makes an important early contribution in understanding those shape dynamics. Unfortunately, I would have liked to see more details about these dynamics and perhaps some effort at quantifying what is happening in some simple and understandable cases.

Response: We thank the reviewer for the thoughtful comment and suggestion. According to the reviewer’s suggestion, we added more details about the dynamics of differential growth-induced 3D structures (Line 258–319). In particular, we quantified the dynamic shape change of the spherical cap structure and compared it with our theoretical model (Line 297–319). We believe that the new results on the dynamic behavior of growth-induced 3D structures significantly improve our manuscript.

More specifically, we quantified the dynamic shapes of the spherical cap structure (shown in Fig. 5a) by measuring (i) Gaussian curvature of the spherical cap-like shape in the center K_{sc} (Fig. 5f), (ii) the location of the shape transition between the spherical cap-like shape ($K > 0$) and the wrinkles ($K < 0$) $(\rho/R_s)_{tr}$ (Fig. 5g, Supplementary Fig. 18, 19), and (iii) the number and amplitude of wrinkles (Fig. 5h, Supplementary Fig. 20). The experimentally measured K_{sc} and $(\rho/R_s)_{tr}$ show a good agreement with those obtained from our theoretical model (dynamic growth function Ω_t). In addition, our theoretical model describes how the number of the wrinkles decreases with time while their amplitude increases, as observed in Enneper’s surfaces.

We added the following figures to show the results: Fig. 5f–h, Supplementary Fig. 18–20.

Comment 4. Finally, however, what the authors do with programming shape dynamics at the end of the paper is truly awe-inspiring. I think this is the most important part of the paper and the greatest advance. I would prefer to see more details about how this works as I found the dynamic programming methods rather confusing as presented.

Response: We highly appreciate the reviewer’s encouraging comment and suggestion. According to the reviewer’s comment, we added more detailed description on how we program 3D structures with sequential motions in the section of “Dynamic 3D structures with programmed sequential motions.”

(1) We added more details in Line 338–340: “To create these structures, we designed Ω_{fast} and Ω_{slow} with the same functional form (Ω for a saddle shape shown in Fig. 2g) but in different Ω ranges: $\Omega_{\text{slow}}/\Omega_{\text{fast}} = C$, where C is a constant and $C > 1$ (Fig. 6c).”

(2) We added the following sentences in Line 340–344: “The growth (swelling and shrinking) rates decrease with t_{ex} and thus with Ω (Fig. 1d, Supplementary Fig. 23), but a 3D shape is determined by the relative growth (not by the absolute values of Ω). We can therefore program the speed of shape transformation without changing 3D shapes by controlling the range of Ω (e.g., the value of Ω_{max} or Ω_{min}) but maintaining the relative growth (e.g., $\Omega(r/R)/\Omega_{\text{min}}$).”

(3) We added Supplementary Fig. 23 that shows the areal swelling rates (speeds of shape change) as a function of Ω .

Comment 5. In the end, I like the paper and I think it should be published. But I would prefer there to be less repetition of what was already done with programmed shape and more detail on the dynamical programming and shape dynamics in general. The basic shape programming can still go in the supplement as it is important to verify that everything works as expected and reproduce existing results.

Response: We thank the reviewer for the comment and suggestion. As the reviewer suggested, we included more details and analysis on the dynamic programming and shape dynamics and shortened the sections related to the basic shape programming.

(1) We restructured and added more details in the section of “Dynamic behavior of growth-induced 3D structures” (Line 258–319).

(2) We deleted the sections (Line 136–141 and 170–180) related to the basic shape programming and the theoretical model, as they are described in details in the Supplementary Information. As Figure 2 shows how 3D structures transform between their shapes at the swelled and shrunk states, we keep Figure 2 in the main text.

(3) We added the following figures relevant to the dynamic programming and shape dynamics: Fig. 5f–h, Supplementary Fig. 17–20, and Supplementary Fig. 23.

Some nitpicks:

Comment 6. In line 33, growth and shrinkage does not define a shape uniquely but only defines the Gaussian curvature uniquely. There are, generically, many shapes with the same Gaussian curvature for topological disks. I think the authors know this and meant something else.

Response: We thank the reviewer for the comment. To make it clearer, we rewrote the sentence in Line 34–36: “This approach defines 3D shapes with Gaussian curvatures^{21,22} and is uniquely capable of creating 3D structures with curved geometries, often seen in biological organisms but difficult to achieve by other methods^{4,5}.”

Comment 7. Line 46–47, I do not think it is fair to the literature to say that mostly axisymmetric shapes are understood. There are several experimental systems with non-axisymmetric shapes

and metrics available and, though there are things still to understand, this sentence seems too strong.

Response: We agree with the reviewer that there are experimental systems with nonaxisymmetric shapes (e.g., as demonstrated in Ref. 4). According to the reviewer's comment, we rewrote the sentence in Line 48: "Furthermore, the principle has been demonstrated for various 3D shapes,^{3,4,36} but achieving nonaxisymmetric 3D structures with complex morphologies remains to be further studied³³⁻³⁵."

Comment 8. Line 157, there is a typo in the word "shape."

Response: We thank the reviewer for carefully reading our manuscript. We corrected the typing error.

Reviewer #2 (Remarks to the Author):

Comment 1. The authors describe some interesting curving and folding results with PEGDA and NIPAM based gels. PEGDA and NIPAM-BIS gels. The tunability of the network is nice and the results are interesting. However, the authors need to more clearly incorporate prior work and correctly reference prior art.

Response: We thank the reviewer for the comment and suggestion. As the reviewer suggested, we included new references in the revised manuscript as described below.

Comment 2. For example, bending and curving by phototunability during crosslinking in a polymer has been discussed previously to create a variety of shapes. (Jamal et al, Differentially photo-crosslinked polymers enable self-assembling microfluidics. *Nature communications*, 2, p.527, 2011).

Response: We agree with the reviewer that this work (Jamal et al., *Nature Communications*, 2011) is one of the key studies in shape-changing 3D structures (based on bending and curving). As suggested by the reviewer, we added this study as a reference (Ref. 11) and cited it throughout the revised manuscript.

Comment 3. Then tuning curvature of folding using PEGDA gels of different molecular weight has also been utilized previously. (Jamal et al, Bio-Origami Hydrogel Scaffolds Composed of Photocrosslinked PEG Bilayers." *Advanced healthcare materials* 2, no. 8 (2013): 1142-1150).

Response: We appreciate the reviewer's suggestion. We included this study (Jamal et al., *Adv. Healthcare Mater.*, 2013) as a reference (Ref. 7) and cited it throughout the revised manuscript.

Comment 4. Finally stating that programmability of shape by diffusion and extent of cross-linking has not been previously achieved is a bit exaggerated. Please see: Cangialosi, A., et al, 2017. DNA sequence-directed shape change of photopatterned hydrogels via high-degree swelling. *Science*, 357(6356), pp.1126-1130.

Response: We agree with the reviewer that the study by Cangialosi and coworkers (Ref. 8: Cangialosi et al., *Science*, 2017) demonstrated DNA sequence-directed shape change of

photopatterned hydrogels, including control of the swelling rates of hydrogel structures using DNA molecules with different sequences (e.g., different hairpin toehold lengths) and hydrogels with different thicknesses. We took great inspiration from this study (Ref. 8) to build dynamic 3D structures with programmed sequential motions (shown in Fig. 6). We therefore cited this study (Ref. 8) several times in the manuscript (5 times in the original manuscript and 6 times in the revised manuscript).

(1) To further clarify the point that the reviewer commented, we rewrote the section relevant to the comment in the revised manuscript. We added a phrase with Ref. 8 in Line 71–76: “The ability to spatially control the rates of shape changes allows us to fabricate dynamic 3D structures with programmed sequential motions, as previously demonstrated with photopatterned hydrogels responsive to different molecular inputs (e.g., DNA molecules with different sequences) and those with different thicknesses⁸. Such ability is critical for implementing complex functions but challenging to attain with global external stimuli (e.g., temperature)⁸.”

As the reviewer is well aware, one of the novel aspects of our study is to achieve programmed sequential motions using global external stimuli (i.e., temperature in our study), difficult to attain via global cues as stated in Ref. 8.

(2) To clearly state the aspect that the reviewer commented and to confine our study to differential in-plane growth-induced 3D shaping (non-Euclidean plates), we rewrote the sentence in Line 43: “However, dynamic growth-induced 3D motions of non-Euclidean plates remains largely unexplored^{3,4}.”

(3) We deleted the following sentence in Line 47–48: “A theoretical framework for predicting and programming dynamic shape changes is absent.”

Comment 5. Finally, biological organisms do not utilize light to program their shape dynamically, so the link to that is a bit tenuous.

Response: We thank the reviewer for this comment. We carefully thought about the link again. We agree with the reviewer that biological organisms do not utilize light to program their shape dynamically.

A key motivation of our study is to use “spatially controlled in-plane growth (swelling and shrinking)” of stimuli-responsive hydrogel sheets to create 3D structures with programmed 3D shapes and motions, as observed in biological organisms. Biological organisms, such as plants and marine invertebrates, often use spatially controlled expansion and contraction of soft tissues (often in a form of thin sheets) to produce complex 3D morphologies and movements (in response to internal signals or environmental stimuli such as temperature, light, and humidity), as discussed in Ref. 3–5 and 23–33 (Line 36–39). Such approaches (e.g., differential growth) have inspired us to create dynamic 3D structures with bioinspired morphologies and motions using spatially-controlled growth of stimuli (temperature)-responsive hydrogel sheets. Although this link can be a bit tenuous, we think that stating the analogy of our approach to how biological organisms (e.g., plants and marine invertebrates) produce 3D shapes and motions could help show the key motivation and biologically inspired aspect of this study and potentially open up new research opportunities.

Comment 6. I recommend that the authors tone down some of the text, include references to prior work accurately and certainly the paper can be interesting from a mechanics perspective.

Response: We thank the reviewer for the comment and suggestion. As suggested by the reviewer, we toned down some of the text in the revised manuscript (especially in the introduction and conclusion) and included references to prior work (Ref. 7 and 11).

Reviewer #3 (Remarks to the Author):

Comment 1. This paper is difficult to read because it is too full of hype. From what can be understood from the figures (and repeated re-reading), the authors control the photo-polymerization of the gel and then once the gel is fabricated, use temperature to control the swelling of the sample. This is a nice notion, but should be explicitly stated as the work. Rather, there is this confusing notion of "growth". They say the word "growth" means expansion and contraction, yet there is not a consistent use of the word in the paper. Does growth also apply to the polymerization phase?

Response: We thank the reviewer for the constructive comments. The reviewer's comments helped us to make the manuscript easier to read and understand.

(1) We agree with the reviewer that the notion of "growth" (expansion and contraction) can be confusing, partly because the word "growth" is also used to describe the growth of polymer chains (e.g., in Ref. 39: Zhou & Johnson, *Angew. Chem., Int. Ed.*, 2013; Ref. 40: Chen et al., *Chem. Rev.*, 2016; Ref. 41: Pan, X. et al. *J. Am. Chem. Soc.*, 2016; Ref. 42: Matyjaszewski, *Adv. Mater.*, 2018.). To clarify the notion of "growth" (expansion and contraction) in this paper, we added quotation marks when we use the word "growth" for the first time in the main text (line 30).

Line 30–32: "A promising approach in this regard is to use spatially controlled in-plane "growth" (expansion and contraction) of hydrogel sheets to form 3D structures via out-of-plane deformation (non-Euclidean plates)^{3,4}."

We consistently used the word "growth" to describe expansion and contraction of soft materials, including biological soft tissues and hydrogels in our manuscript. The word "growth" has been used to describe the expansion and contraction of hydrogels and other soft materials in differential "growth" (expansion and contraction)-induced 3D shaping (non-Euclidean plates) (e.g., Ref. 4: Kim et al., *Science*, 2012, Ref 33: van Rees et al., *Proc. Natl. Acad. Sci. U. S. A.*, 2017). We believe that the clarification made in the revised manuscript would make it easier to read and understand.

(2) Growth does not apply to the polymerization phase in our manuscript. To make it clear, we added "swelling and shrinking" after growth when we describe the polymerization process in Line 88–90: "The phototunability provides a flexible means to encode hydrogels with spatially and temporally controlled growth (swelling and shrinking), which can be used to program the formation of 3D structures and their motions."

(3) We agree with the reviewer that it can be difficult to read the manuscript, partly because of the notion of growth and the concept of how we make 3D structures by spatially controlled growth (swelling or shrinking) of hydrogel sheets (non-Euclidean plates). As the basic concept of differential growth-induced 3D shaping (non-Euclidean plates) is described in the key references (e.g., Ref. 3: Klein et al., *Science*, 2007; Ref. 4: Kim et al., *Science*, 2012), we described the detailed theoretical background and our theoretical model in Supplementary Information (rather than in the main text). To make the manuscript easier to read, we added the following sentence after describing the general concept in the introduction of the revised manuscript (Line 32–34): “Because bending is energetically less expensive than stretching in a thin sheet, the internal stresses developed by nonuniform in-plane growth are released by out-of-plane bending deformation^{3,4}.”

(4) In addition, we rewrote and toned down the text in the revised manuscript (especially in the introduction and conclusion) to make it easier to read and understand. We also restructured and added more details in the sections of “Dynamic behavior of growth-induced 3D structures” (Line 258–319) and “Dynamic 3D structures with programmed sequential motions” (Line 338–339) to make the manuscript easier to read.

Comment 2. The work is interesting, though it is not clear that it belongs in Nat. Comm. It may be better for a more specialized journal. Even if it goes to a different journal, the authors should think about rewriting the introduction and more clearly stating what exactly they have done—as I understand it, it is sufficiently interesting that they program the photo-polymerization to control the 3D shape of thermo-responsive gels. In this context, it is worth citing work by Jeremiah Johnson at MIT and K. Matyjaszewski at CMU and then discussing how the new work is different from the latter studies involving photo-polymerization.

Response: We thank the reviewer for the constructive comments and suggestions.

(1) According to the reviewer’s comments, we rewrote some parts of the introduction and toned down the text in the revised manuscript, including those described in our responses to Comment 1 of the reviewer.

(2) We appreciate the reviewer’s suggestion for citing and discussing the work on controlled photopolymerization by Dr. Johnson at MIT and Dr. Matyjaszewski at CMU. We believe that the discussion of state-of-the-art controlled/living radical photopolymerization (CRP) makes the manuscript more interesting and potentially opens up new research opportunities.

As suggested by the reviewer, we compared our photopolymerization approach with the photoinduced controlled/living radical polymerization (photo-CRP) with relevant references (Ref. 39–42) in Line: 113–117: “In contrast to photoinduced controlled/living radical polymerization (photo-CRP), which precisely controls the molecular architecture of polymers, such as molecular weights and compositions,³⁹⁻⁴² our approach modulates only the overall density of polymer networks (rather than the molecular weight of individual polymer chains).”

We added the following references in the revised manuscript: Ref. 39: Zhou & Johnson, *Angew. Chem., Int. Ed.*, 2013; Ref. 40: Chen et al., *Chem. Rev.*, 2016; Ref. 41: Pan, X. *et al. J. Am. Chem. Soc.*, 2016; Ref. 42: Matyjaszewski, *Adv. Mater.*, 2018.

REVIEWERS' COMMENTS:

Reviewer #1 (Remarks to the Author):

The current version of the paper is much improved. It seems the authors made extensive revisions in response to all the referee comments, substantially toned down the hype, and managed to better explain their accomplishments and results.

I am happy with the current version of the paper. I do think it could be published in Nature Communications.

Reviewer #2 (Remarks to the Author):

The authors have satisfactorily addressed issues and the paper can be accepted

Reviewer #3 (Remarks to the Author):

The authors have successfully answered my questions.

But now that I better understand their study, I believe the authors should refer to the following paper, which deals with designing photo-responsive gels that can be shaped into different geometries and driven to move:

Kuksenok, O. and Balazs, A.C., "Modeling the Photo-induced Reconfiguration and Directed Motion of Polymer Gels" Adv. Func. Mater., 23 (2013) 4601-4610.

Point-by-point Response to the Reviewers' Comments

We thank the reviewers again for their constructive and thoughtful comments. According to the reviewers' comments, we revised our manuscript as described below.

Reviewer #1 (Remarks to the Author):

Comment. The current version of the paper is much improved. It seems the authors made extensive revisions in response to all the referee comments, substantially toned down the hype, and managed to better explain their accomplishments and results. I am happy with the current version of the paper. I do think it could be published in Nature Communications.

Response: We appreciate the reviewer's comment.

Reviewer #2 (Remarks to the Author):

Comment. The authors have satisfactorily addressed issues and the paper can be accepted

Response: We appreciate the reviewer's comment.

Reviewer #3 (Remarks to the Author):

Comment. The authors have successfully answered my questions. But now that I better understand their study, I believe the authors should refer to the following paper, which deals with designing photo-responsive gels that can be shaped into different geometries and driven to move: Kuksenok, O. and Balazs, A.C., "Modeling the Photo-induced Reconfiguration and Directed Motion of Polymer Gels" *Adv. Func. Mater.*, 23 (2013) 4601-4610.

Response: We thank the reviewer for the comment and suggestion. We included the study (Kuksenok and Balazs, *Adv. Funct. Mater.*, 2013) as a reference (Ref. 13) and cited it in the introduction of the revised manuscript.